# Light environment drives evolution of color vision genes in butterflies and moths

Yash Sondhi[1✉], Emily A. Ellis [2], Seth M. Bybee[3], Jamie C. Theobald [1] & Akito Y. Kawahara [2]

Opsins, combined with a chromophore, are the primary light-sensing molecules in animals and are crucial for color vision. Throughout animal evolution, duplications and losses of opsin proteins are common, but it is unclear what is driving these gains and losses. Light availability is implicated, and dim environments are often associated with low opsin diversity and loss. Correlations between high opsin diversity and bright environments, however, are tenuous. To test if increased light availability is associated with opsin diversification, we examined diel niche and identified opsins using transcriptomes and genomes of 175 butterflies and moths (Lepidoptera). We found 14 independent opsin duplications associated with bright environments. Estimating their rates of evolution revealed that opsins from diurnal taxa evolve faster —at least 13 amino acids were identified with higher dN/dS rates, with a subset close enough to the chromophore to tune the opsin. These results demonstrate that high light availability increases opsin diversity and evolution rate in Lepidoptera.

[1] Department of Biology, Florida International University, Miami, FL, USA. [2] McGuire Center for Lepidoptera and Biodiversity, Florida Museum of Natural History, University of Florida, Gainesville, FL, USA. [3] Department of Biology, Brigham Young University, Provo, UT, USA. ✉email: ysond001@fiu.edu

The evolution of light detection has allowed animals to monitor ambient brightness for circadian rhythms, determine the intensity of light from different directions for phototaxis, and form images for proper spatial vision[1]. With images, animals track and correct their own motion, target or avoid objects, and sometimes infer more elusive properties of a scene, such as object nearness or time until a collision. Although some animals achieve spatial vision without dedicated structures, like the diffuse photoreception of sea urchins[2], overwhelmingly, acuity is captured by eyes, the diverse and convergently evolved organs that arrange screening pigments, optical lenses, and photoreceptors, to focus and capture images[3]. When light is sufficient, some estimate of its wavelength composition can improve object discrimination even under diverse or uneven lighting[4]. This perception of color requires photoreceptors with different wavelength sensitivities whose signals are then compared by underlying opponent processes[5].

Wavelength sensitivities result largely (but not wholly) from a photoreceptor's visual pigments, which are formed by joining a light-sensitive, retinal-based chromophore to an opsin, a seven-transmembrane-domain G protein-coupled receptor protein. When a chromophore responds to a photon absorption that changes its conformation, the opsin transduces this into a biologically meaningful signal by activating an internal G protein signaling cascade. Large-scale phylogenomic analyses have found many duplications and losses in the opsin protein family across invertebrates[6,7]. The water flea, *Daphnia pulex* possesses 46 opsins[8], dragonflies have 15–33 opsins[9–11] and mantis shrimp have 12–33 opsins[12]. However, any more than four spectral channels offer diminishing returns in extracting information from natural scenes[13,14], so why do some animals have so many opsins?

Since they regulate visual light transduction, opsins are subject to strong adaptive evolution, but they can also evolve through non-adaptive mechanisms. Non-adaptive forces usually cause random sequence evolution, and unless the duplicated opsins are co-opted for visual use, they usually become pseudogenes. Adaptive evolutionary forces are more likely to cause consistent, repeated, and persistent patterns of opsin retention and diversification[15], such as with mate choice in guppies and butterflies[16,17], flower foraging in bees and wasps[18] and changing light intensity environment with many nocturnal animals[19,20].

Sensory modalities such as smell, electromagnetic reception, and touch can be more reliable than vision in dim environments[19]. Resource allocation trade-offs can cause a loss of stabilizing selection on genes of inefficient sensory systems, resulting in their downregulation or them becoming pseudogenes. This has been seen in nocturnal mammals[21], cave-dwelling crayfish[22], and deep-sea organisms[23–25]. If diminished light availability causes reduced opsin expression and loss, abundant light, conversely, may cause higher opsin expression, or prevent the loss of duplicated opsins and eventually lead to functional divergence. A comparison of visual genes between diurnal and nocturnal Lepidoptera revealed elevated opsin expression in the diurnal species[26]. Similarly, a study of opsin evolution across fireflies found higher amino acid transition rates in diurnal fireflies, across four independent diel switches[27].

Lepidoptera opsin diversity has been studied in a handful of model taxa—mostly diurnal butterflies[26,28,29]—but has yet to be studied comprehensively across the entire order and multiple diel niches. Opsins are characterized by the wavelength to which their response is maximum ($\lambda_{max}$). The $\lambda_{max}$ is sufficient to approximate the response curve of most opsins[30]. Lepidoptera opsins are usually classified as UV/RH4, Blue/RH5, and LW/RH6 opsins with corresponding maximal responses ($\lambda_{max}$) in the ultraviolet (UV) (300–400 nm), blue (400–550 nm), and green/red (450–620

nm) wavelengths (LW). RH4–6 are implicated in color vision, but Lepidoptera also possess the non-visual RH7, which is associated with light sensing needed to maintain circadian rhythm[31].

Earlier studies on Lepidoptera opsin evolution include work by Briscoe[28], who analyzed the visual genes of eight Lepidoptera species including two moths, Xu et al.[32], who analyzed 30 species including 12 moths, and Feuda et al.[31] who analyzed 10 species including four moths. However, due to small sample sizes, these studies had limited statistical power. They used gene trees instead of species trees for selection analyses, which, if different from the species tree topology, can bias results[33]. The few studies that examined opsin diversity and diel-niche association[31,32] compare butterflies and moths, effectively using only a single diel switch. But Lepidoptera have more than 100 recorded diel transitions[34], and only by examining multiple independent diel-niche switches can we understand how light environment and diel-niche drive the evolution of their visual systems. To test if bright environments drive opsin diversification, we mine genomes and transcriptomes of 175 Lepidoptera species for visual opsins, combine our annotations with natural history diel-niche from the literature and map these traits onto a well resolved tree[35] to examine their evolution.

## Results

**Lepidoptera opsin duplications associated with bright environments: mapping opsin diversity and diel-niche.** We examined patterns of opsin diversity across Lepidoptera by mining assembled transcriptomes[35] and annotated genomes from Lepbase[36] and Ensembl Metazoa[37] (Supplementary Data 1). Transcriptome quality was assessed using BUSCO[39] and visual score, calculated as the percentage of genes recovered from a well characterized set of Lepidoptera visual genes[26] (Supplementary Data 1,2). A phylogenetically informed annotation approach (PIA)[40] was used for opsin annotation (Supplementary Data 1), which we reconfirmed by building nucleotide and amino acid gene trees (Supplementary Figs. 1, 2). We recovered at least one opsin from 114 of 175 Lepidoptera taxa (Supplementary Data 1) and were able to confirm diel-niche for 104 of these species using the literature, in consultation with experts.

We mapped opsin diversity—specifically UV/RH4, Blue/RH5, LW/RH6, and RH7 opsin recovery—and diel-niche onto a well-resolved Lepidoptera phylogeny[35] (Fig. 1). Species with duplications were counted (Table 1) and the number of duplication events were estimated using a tree reconciliation analysis (Supplementary Data 3, Supplementary Figs. 3–5). Chi-squared tests determined if the number of diurnal species with duplications was more than what could be expected by chance. Duplication events occurred more often in diurnal lineages ($\chi^2 = 6.025$, $p$-value = 0.014, Table 1) and increased duplication in diurnal species was evident even after we excluded species with ambiguity in diel-niche assignment ($\chi^2 = 8.478$. $p$-value = 0.0035, Table 1).

Transcriptome quality was variable (Supplementary Data 1, 2) and to ensure the trends were not an artifact of this variation, we investigated whether transcriptome quality was correlated to phylogeny and found no evidence for such a correlation (BUSCO, $K = 0.3143298$, $p$-value = 0.151; visual score, $K = 0.3846586$, $p$-value = 0.071). To test if transcriptome quality was correlated with diel-niche after accounting for the effect of phylogeny, a Phylogenetic ANOVA was performed, but it failed to reveal any correlations (BUSCO, $F = 0.375$, d.f. = 4, $p$-value = 0.8256; visual score, $F = 0.25$, d.f. = 4, $p$-value = 0.8213).

We performed an ancestral state reconstruction (ASR) of diel-niche, each opsin and total number of opsins. Opsin losses are notoriously difficult to confirm[31] and while both tree reconciliation

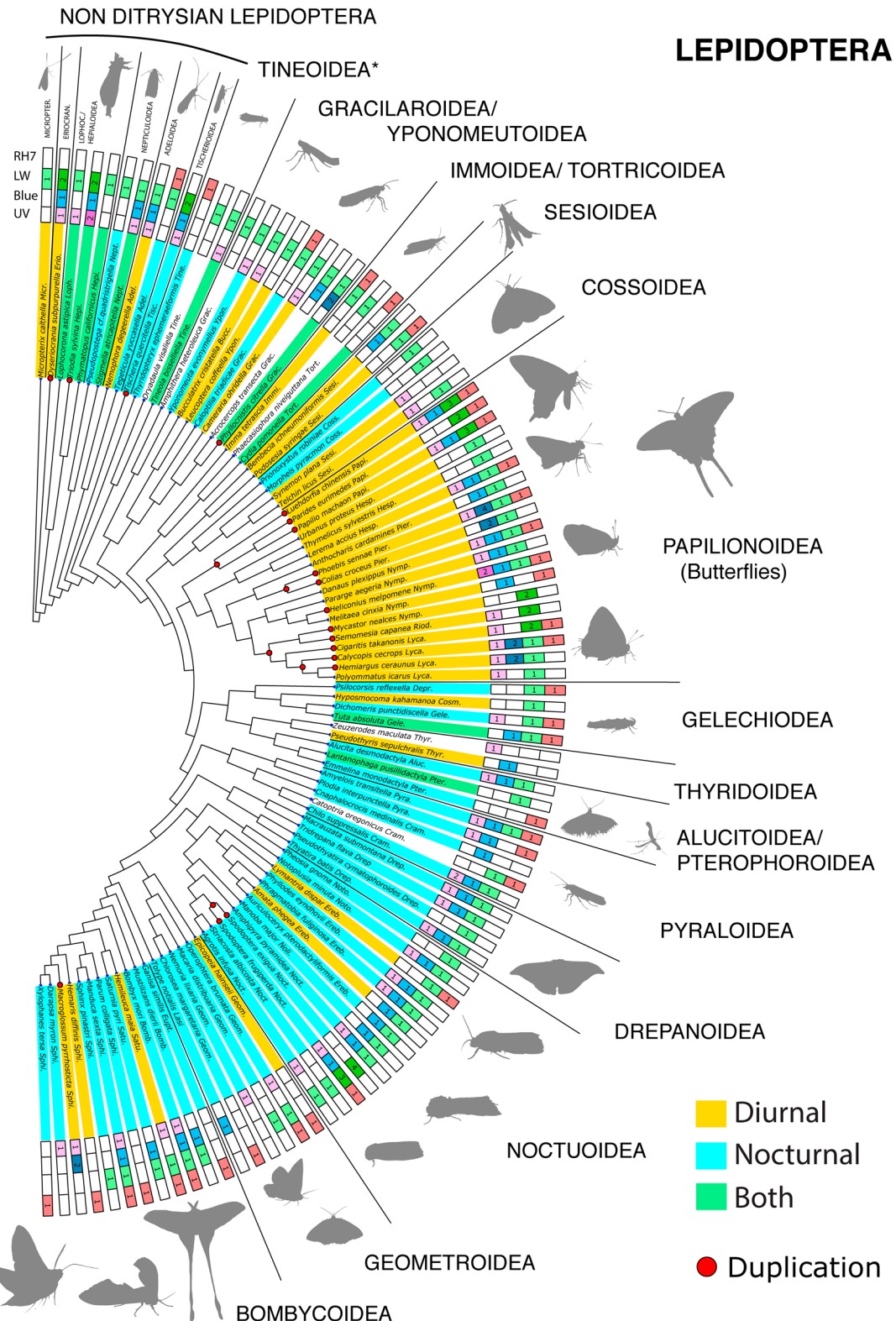

**Fig. 1 Opsin annotation and diel activity are mapped onto a Lepidoptera species tree. Duplications are associated with taxa active in bright light.** Taxa are color coded by diel-niche and RH4/UV, RH5/Blue, RH6/LW, and RH7 opsin recovery is marked. Red dots at nodes indicate duplication events identified by the tree reconciliation analysis for that lineage and darkened colors indicate a duplication in a particular opsin. Duplications (red dots) are more commonly associated with diurnal (yellow) and and partially diurnal or"both" taxa (green) which are more active in bright light environments. The tree is a pruned cladogram of tree of the most recent Lepidoptera phylogeny[35]. 98/104 taxa are included in the figure, taxa for which the identification were only till genus level are excluded, but all the species with duplications are shown. Asterisk (*) indicates that these superfamilies are not monophyletic. Family names and some superfamily names are abbreviated, see Supplementary Data 1 for expanded names and complete classification, raw data data used to create these figures can be found at ref. [38].

**Table 1 Chi-square values for nocturnal vs. diurnal Lepidoptera species, including and excluding crepuscular and "both" species.**

| Dataset | Chi-square coefficient | p-value | No. of samples D:N | No. of duplications D:N |
|---|---|---|---|---|
| nocturnal vs. diurnal + crepuscular + both | 10.11 | 0.0014 | 51:54 | 16:3 |
| nocturnal vs. diurnal + crepuscular + both[#] | 7.97 | 0.0047 | 51:46 | 16:3 |
| diurnal vs. nocturnal + crepuscular + both | 9.32 | 0.0022 | 42:63 | 14:5 |
| diurnal vs. nocturnal + crepuscular + both[#] | 7.41 | 0.0064 | 42:55 | 14:5 |
| nocturnal vs. diurnal | 10.67 | 0.001 | 42:54 | 14:3 |
| nocturnal vs. diurnal[#] | 8.478 | 0.0035 | 42:46 | 14:3 |
| nocturnal vs. diurnal[**] | 6.0259 | 0.014 | 42:46 | 14:5 |
| nocturnal vs. diurnal[**] (yates-correction) | 4.8171 | 0.02818 | 42:46 | 14:5 |

Rows 1–7 is count data of the species with identified opsins. Row 8–9 is the number of duplication events from the tree reconciliation analysis. Duplications occur at a higher frequency in diurnal species than expected by chance, even after accounting for various kinds of uncertainty in diel niche assignment. (All results are significant $p < 0.05$).
 D diurnal, N nocturnal.
#excluding uncertain diel states.
**From tree reconciliation analyses.

and ASR results are included (Supplementary Figs. 3–10)., we refrain from interpreting the losses.

*Ultraviolet (UV).* 50% (57/114) of the taxa recovered UV/RH4 opsins (Fig. 1, Supplementary Figs. 3, 7). UV opsins had duplications in only 3 independent lineages; the diurnal *Heliconius melpomene*, in which UV opsin duplication has been recorded before[41] and crepuscular *Triodia sylvina* (Hepialidae), an ancient lineage of ghost moths, known for swarming at dusk[42]. The only nocturnal species with a UV duplication was *Chilo supressalis*, an important pest species[43].

*Blue.* 46.4% (53/114) of the taxa recovered Blue/RH5 opsins (Fig. 1, Supplementary Figs. 4, 8). Blue opsin duplications were present in 5 families, with 6 different duplication events, all occurring in diurnal species. The pierid species, *Colias croceus* and *Phoebis sennae*, and two lycaenid genera, *Calycopis* and *Hemiargus*, also have Blue opsin duplications. Behavioral and electrophysiological data have shown that these families, if not these individual species, have functional duplications[28,44,45]. *Macroglossum pyrrhosticta*, a diurnal hawkmoth (Sphingidae), also had a Blue duplication.

*Long wavelength (LW).* 83.3% (96/114) of taxa recovered LW/RH6 opsins, the highest capture rate of all three opsin families (Fig. 1, Supplementary Figs. 5, 9). LW opsins duplications were recorded in 8 families and 11 genera, with 10 duplication events. A Mann–Whitney U test indicated that there were more LW opsins in diurnal taxa than nocturnal taxa ($U = 662.0$, p-value = 0.033). Only two of these genera with duplications are nocturnal. We recovered previously reported LW duplications in one Lycaenidae, two Riodinidae, and three Papilionidae species[28]. Butterflies are among the few insects known to have sensitivity to the color red, through LW duplication or filtering pigments[28,46]. Thus, duplications may indicate true color expansion. Other diurnal or crepuscular species with LW duplication were *Dyseriocrania subpurpurella* (Eriocraniidae) and *Triodia sylvina* (Hepialidae).

The tiger moth *Callimorpha dominula* (Erebidae: Arctiinae) and the giant butterfly moth *Paysandisia archon* (Castniidae), two diurnal moths not in this dataset, also have LW duplications, and *Paysandisia* is known to have sensitivity to red wavelength light[31,44]. *Spodoptera* (Noctuidae), of which some species are invasive pests, is nocturnal but has a red sensitive LW duplication. Many *Spodoptera* species are migratory, and flying above the clouds of the night sky may free them from low light constraints[47,48]. *Tischeria quercitella* (Tischeriidae), a leaf mining

moth, is the only other nocturnal species found with a LW duplication, despite examining over 50 nocturnal species. In addition to the visual opsins, the non-visual RH7 opsin was also analyzed (Supplementary Fig. 10), but we recovered far fewer opsins with only 28% (33/114) of taxa recovering RH7 opsins. No duplications were found, consistent with other work[31].

**Diurnal species opsins have higher selection rates than nocturnal species: Opsin selection in Lepidoptera.** PAML estimated rates of selection (ω or dN/dS) and we tested if these rates differed between nocturnal and diurnal Lepidoptera. For datasets that showed significant differences, branch-site models in PAML and HyPhy were used to identify amino acids under selection (Supplementary Data 4). Sensitivity to different sample sizes and starting trees was tested (Fig. 2, Table 2, Supplementary Data 4). To ensure comparison of opsin rates across the same group of species, analyses were limited to species that recovered all three visual opsins. We were unable to use the entire dataset of recovered opsins for UV, Blue, and LW opsins because of low quality alignments and manually trimmed or removed sequences that reduced the alignment quality (see "Methods" for details).

We first ran PAML on a subset of the data ($n = 12$–$14$), similar in size to those used by Feuda et al.[31] and Xu et al.[32], to determine the effect of sample size, although taxon sampling across datasets was non-uniform. Of all three visual opsins, only UV opsin rates were a significantly better fit when partitioned into two rates, using diel niche, than one rate for all branches (p-value = <0.001, $\omega_{diurnal(d)} = 0.0447$, $\omega_{nocturnal(n)} = 0.0234$) (Fig. 2A, Table 2, Supplementary Data 4).

Next, using larger sample sizes and similar species across datasets ($n = 24$–$27$), we found significant differences between diel-niche for all three color opsins. The dN/dS rates were higher in diurnal species than nocturnal species. UV (p-value <0.001, $\omega_d = 0.04816$, $\omega_n = 0.01643$) and LW opsins (p-value <0.001, $\omega_d = 0.052$, $\omega_n = 0.0117$) had a greater magnitude of differences than Blue opsins (Fig. 2B, Table 2, Table S4). RH7 acted as a control because it is not involved in vision and had no significant differences for rates between diel-niches.

Since Blue and UV opsin recovery was sparse (~50% of the taxa) and gene tree topology might be biased, we ran the PAML analysis using a species tree instead of gene trees. The species tree models showed significant differences between diel-niches for all three visual opsins (p-value <0.001) (Fig. 2C, Table 2, Supplementary Data 4). RH7 was excluded from this analysis in order to compare across similar species trees, which was precluded by the poor overlap of species between RH7 and visual opsins.

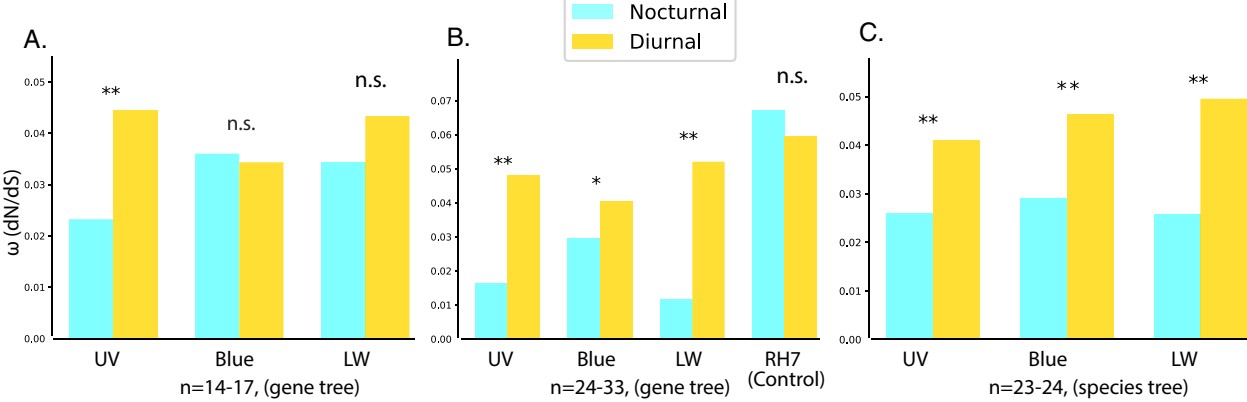

**Fig. 2 Opsin selection (dN/dS) rates between nocturnal and diurnal Lepidoptera species. Different models show that rates for visual opsins are higher in diurnal taxa. A** Blue, LW ($n = 17$ taxa) and UV ($n = 14$ taxa) opsin dataset were run using opsin gene trees. **B** Expanded dataset ($n = 24$ taxa) for UV, Blue, and LW opsins run using gene trees. RH7 was analyzed using more species ($n = 33$ taxa). **C** Expanded dataset run using a robust species trees (UV, LW $n = 24$ taxa, Blue = 23 taxa). *$p$-value < 0.05, **$p$-value < 0.01. The significance values indicate how well different models fit the data using likelihood ratio test (LRT). Model parameters used to run PAML and estimate the rates represented in the figures can be found at ref. [38]. Opsin selection rates modeled for various datasets with different sample sizes and different model parameters all show that RH4/UV, RH5/Blue, and RH6/LW visual opsins have higher dN/dS rates in diurnal species than nocturnal species. RH7, a non-visual opsin, shows no significant difference. Smaller sample sizes (**A**) often fail to detect differences, which were more apparent when using larger data sets (**B**). Using a robust species tree rather than the gene tree shows more consistent differences across the visual opsins (**C**).

**Table 2 dN/dS ($\omega$) rates for null ($M_0$) and branch models ($M_2$) for UV, Blue, Long wavelength (LW), and RH7 opsins using species and gene trees and different number of taxa. Increasing the number of taxa shows significant differences between rates across nocturnal ($\omega_1$) and diurnal species ($\omega_2$) for color opsins but not for RH7 opsin, which served as a control.**

| Opsin | $\omega_0$ | $\omega_1$ | $\omega_2$ | $n$ | $LnL_0$ | $LnL_2$ | d.f. | 2LnL | $p$-value |
|---|---|---|---|---|---|---|---|---|---|
| $M_2$ vs. $M_0$ gene trees | | | | | | | | | |
| UV | 0.0328 | **0.0482** | **0.01643** | 24 | −13222.396 | −13196.06 | 48 | 52.54 | <0.001* |
| Blue | 0.03251 | 0.0403 | 0.02945 | 23 | −11310.806 | −11308.083 | 46 | 5.449 | 0.0196 |
| LW | 0.03264 | **0.0521** | **0.01173** | 24 | −11232.972 | −11174.84 | 48 | 116.27 | <0.001* |
| RH7 | 0.0653 | 0.0594 | 0.06719 | 33 | −22578.67 | −22577.68 | 66 | 1.978 | 0.15 |
| UV | 0.03058 | **0.0445** | **0.02324** | 14 | −6892.501 | −6887.289 | 28 | 9.6 | <0.001* |
| Blue | 0.0354 | 0.0343 | 0.03602 | 17 | −9822.971 | −9822.9126 | 34 | 0.115 | 0.735 |
| LW | 0.03893 | 0.0434 | 0.03445 | 17 | −6690.525 | −6689.4261 | 34 | 2.197 | 0.138 |
| $M_2$ vs. $M_0$ species trees | | | | | | | | | |
| UV | 0.03206 | **0.0411** | **0.02609** | 24 | −13265.664 | −13260.007 | 49 | 11.314 | <0.001* |
| Blue | 0.03361 | **0.0465** | **0.02922** | 23 | −11333.887 | −11328.244 | 47 | 11.287 | <0.001* |
| LW | 0.03309 | **0.0495** | **0.02589** | 24 | −11247.362 | −11233.584 | 49 | 28.779 | <0.001* |

Using species tree shows differences for all three opsins. $\omega_0$: rates for null model, $\omega_1$: rates for diurnal species, $\omega_2$ for nocturnal species, n: number of taxa, $LnL_{0/2}$: log likelihood scores for $M_0/M_2$, d.f.: degrees of freedom for $M_2$. 2LnL: twice the $LnL_2$–$LnL_0$, * highly significant $p$-value (<0.001), with rates from corresponding models in bold.

To test if particular sites were under positive selection in diurnal species, we used site and branch site-models. The site models test if any sites have different dN/dS rates across all branches. The analyses found no significant sites under positive selection across all branches showing that if diel-niche is not considered, there is no underlying signal of elevated dN/dS rates. The branch-site models, which allow partition by diel-niche, did identify amino acid sites that were under a higher dN/dS in diurnal species in UV, Blue, and LW opsins, however it was not able to classify them as being under positive selection. This is expected in opsins because they have a low dN/dS rates overall[49], with $\omega < 0.5$, and the null models fixes dN/dS rates to $\omega = 1$, preventing it from classifying it as significant positive selection. Instead, we used more sensitive tests like the Contrast-FEL[50] and MEME models[51] from HyPhy[52] to verify the branch site models (Supplementary Data 4). These models look for differences in dN/dS rates in different sites among different branches, as opposed to looking for rates greater than one and thus are more

appropriate for opsins. They are more recent implementations and are designed to be more sensitive for detecting selection since they are tailored to detect specific kinds of evolutionary scenarios[50–52]. They showed similar results as the PAML analysis but revealed additional sites under higher dN/dS in diurnal opsins (Supplementary Data 4). There were some opsin specific patterns, UV and LW opsins had considerable overlap in recovered sites for the two different analyses of sites under selection, but Blue opsins had less overlap in the identified sites (Supplementary Data 4). Model sensitivity to input tree also had opsin specific trends. With Blue and LW opsins, sites under selection predicted by the models were unaffected by the choice of gene vs. species trees but it did affect the results of UV opsins (Table S4).

Xu et al.[32] reported elevated dN/dS rates in butterflies (diurnal) compared to moths (nocturnal), with LW, Blue, and UV opsins, showing a decreasing magnitude in differences. In contrast, our study finds UV opsins had the highest and most consistent dN/dS rate differences. Feuda et al.[31] used two independent diurnal

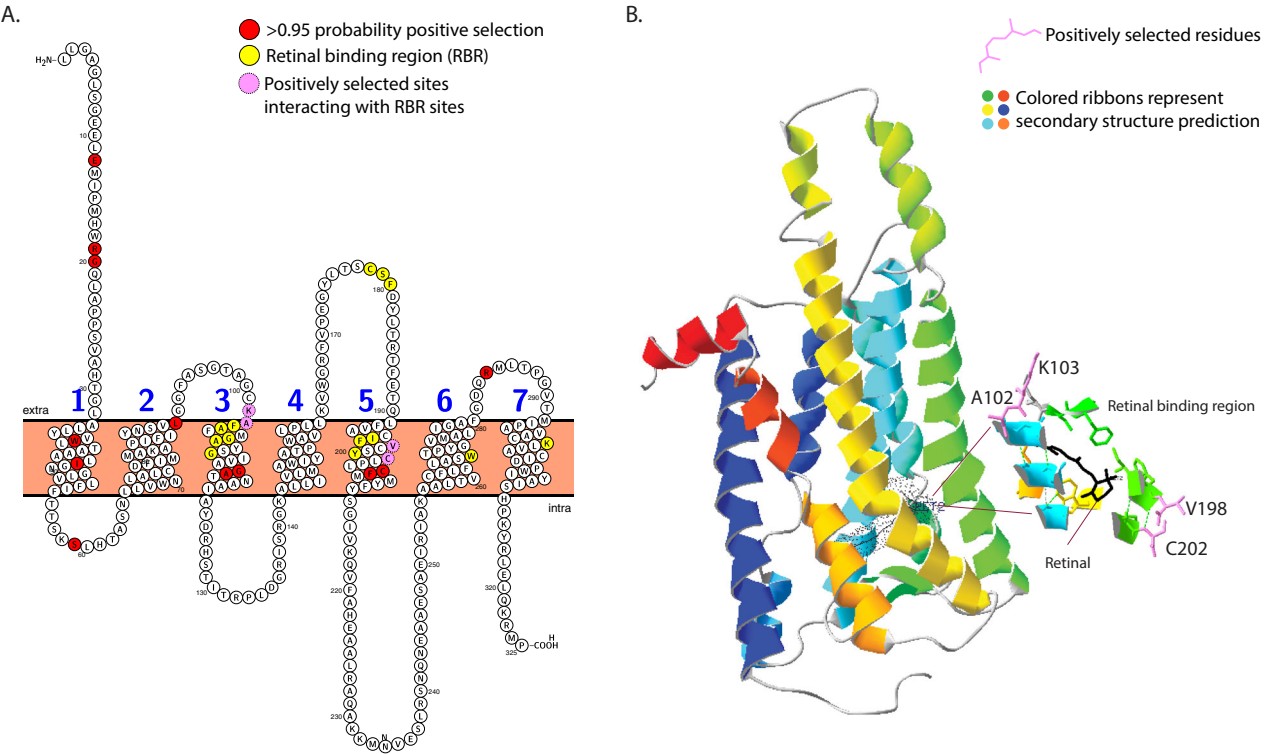

**Fig. 3 Opsin structural prediction with retinal binding sites, sites with higher dN/dS (positively selection) and putative tuning sites, the subset that interacts with both. A** UV-opsin PHOBIUS transmembrane prediction. Opsins sites with higher dN/dS rates and amino acids close to (<=4 Å) retinal, which form the retinal binding region (RBR) have been marked. **B** UV-opsin model made using squid rhodopsin as the template. Retinal is marked, secondary structure prediction of alpha helices are shown in different colors and the expanded view shows the positively selected sites that can interact (<=2 Å) with the retinal binding sites. Protein models used to get these metrics can be found at ref. [38]. Sites under positive selection are highlighted in red, sites in the retinal binding region in yellow and positively selected sites that are capable of interacting with the retinal binding sites in pink, identifying likely candidates for opsin tuning (**A**). Other positively selected are located within the transmembrane region and could be indirectly involved in opsin tuning (**B**).

transitions, with a total of 10 Lepidoptera taxa and have results similar to ours (Fig. 2B). UV and Blue and LW genes in nocturnal Lepidoptera underwent higher dN/dS rates in diurnal species but RH7 had almost similar levels of selection in both nocturnal and diurnal species.

**Expansion of color vision through sequence tuning: mapping sites to predicted protein structure.** We mapped the significant sites with higher selection rates (positive selected sites) onto the predicted protein structure (see "Methods" section) and the transmembrane helix predictions (Fig. 3A, B). Each class of opsin mapped sites to a unique pair of adjacent transmembrane helices (Supplementary Fig. 11). 13 amino acids were identified in the retinal binding region (Fig. 3A, B; Supplementary Data 4) and we mapped these retinal binding sites to transmembrane and 3D structures (Fig. 3A, B), to see if there were any interactions. The positively selected amino acids at sites 102, 103, 198, and 102 are bonded to the retinal binding sites (<2.0 Å) and hence are strong candidates for spectral tuning (Fig. 3, Supplementary Data 4). Eight other positively selected sites also map to transmembrane regions and may tune the opsin indirectly. In addition, UV 59 may be a convergent site of selection, as it has previously been identified in diel-transitions of fireflies[27].

## Discussion
What does opsin duplication or loss mean for the visual system of an organism? One consequence of opsin duplication and divergence is the potential for improved color discrimination. Color

vision usually requires at least two opsins with a partial overlap in spectral sensitivity and even small changes to the amino acid sequence can modify and tune the opsin, shifting the perceived color space of an organism. Alternatively, more dramatic shifts in detectable color space can occur through loss or gain of opsins. For example, many butterflies can see in the red (620–780 nm) due to Long Wavelength (LW) opsin duplication and divergence[29]. RH4–6 are implicated in color vision, but non-visual RH7 opsin is associated with circadian rhythm maintenance and has a phylogenetically scattered distribution among Lepidoptera.

Examining opsin diversity across multiple lineages that have switched diel-niche can help determine how light environment affects opsin evolution. Dragonflies, mosquitoes, and butterflies show multiple opsin duplications with as many as 33 color opsins[10,29,31,46], while beetles and scorpionflies show losses[53,54]. These studies have not examined potential links between diel activity and opsin diversity, or have failed to find consistent trends (reviewed in ref. [31]). Analyses are confounded by uncertainty in diel-niche assignment and a lack of multiple independent diel-switches. Systematic error—including shallow sequencing and poorly resolved species trees—could also obscure any trends[33,55].

The greatest limitation of large-scale gene mining approaches is the reduced power to detect absences. Low coverage transcriptomes from older studies, mixed tissue sources, as well as varied assembly and sequencing methods, all increase heterogeneity in opsin recovery. Ideally, one would sequence only eye tissue with muscle tissue from the body as a control, but complete

information about which tissues generated transcriptomes is often lacking. Further, researchers typically do not measure expression levels, which could identify non-functional opsins[56]. However, because multiple quality control metrics and annotation methods were used here, we believe we have obtained the best possible estimate of opsin diversity with these data. While some transcriptomes had lower quality, variation in quality was randomly distributed across the phylogeny (i.e., phylogenetic signal was low). Furthermore, after accounting for phylogeny, we were unable to find evidence for correlations of diel-niche with transcriptome quality, together indicating that the trends with diel-niche are unlikely to be an artifact of transcriptome quality. We are not confident that the losses we identify are true losses and refrain from making claims about trends for losses but include both the ancestral state reconstruction and tree reconciliation results (Supplementary Figs. 3, 4). Qualitatively, diurnal lineages and nodes appeared to have more opsins. Although we were not able to confirm this quantitively in the reconstructions, we did find the trend of higher diurnal opsin counts from the extant taxa for the better recovered LW opsins.

The selection models implemented in PAML are sensitive to a range of parameter conditions such as choice of tree topology, quality of alignment and sample size. We ran these models using a range of a parameters and presented the results. It is shown that branch length in the tree topology can affect these analysis[33] and therefore, we used the species tree rather than the gene trees models for estimating sites under positive selection. The species tree had better estimates of evolutionary branch length because it was constructed using phylogenomic data[35]. We also used several newer models, implemented in HyPhy, to detect sites under positive selection. These models are tailored to be more sensitive in the detection of specific kinds of evolutionary scenarios[50–52] and unlike the PAML these models do not assume dN/dS rates =>1, but just search for site classes with different rates and thus are better suited opsins, which almost never have any sites with dN/dS rates >=1.

Our analyses show that at least four positively selected sites are in the bio-physical range of direct interaction of the retinal binding pocket and hence could influence opsin spectral tuning in diurnal species by changing the chemical environment surrounding retinal. Feuda et al.[31] also map the positively selected amino acids to opsins[31], but they recover a large fraction of sites in terminal regions, not in the helices. This could be due to alignment methods (i.e., removing gap-filled regions). We refrained from removing gap filled regions from the alignment, instead, we performed end trimming and limited the analysis to species that resulted in a gapless alignment.

More than 75% of the opsin duplications events that we discovered across 10 independent lineages are in diurnal species, which additionally have higher selection (dN/dS) rates than nocturnal species. Differences in selection rate between the diel niches are visible across all three visual opsins, but are most robust in ultraviolet-sensitive opsins. Structural modeling shows that in diurnal species, most amino acids under positive selection are in the transmembrane helices and a subset interact with the retinal binding region.

Each of these observations supports our prediction that transitions from dim to bright-light environments drive opsin diversification in Lepidoptera. An alternative scenario, where a diverse repertoire of opsins allows the species to invade new light environments is also possible. However, ancestral state reconstruction indicates the shift to diurnality occurred prior to the duplication of opsins. However, because deep-time nodes are more difficult to reconstruct, the specific mechanism that gives rise to the results we document are difficult to resolve.

The ancestors of both diurnal and nocturnal lepidopterans were likely crepuscular with limited trichromatic vision[34], and if diel-niche shifts and opsin diversification are linked, we first need to know why their diel-niche switched. One hypothesis that has been proposed is that moths that were nocturnal switched to becoming diurnal to escape predatory bats[57]. Moving to diurnal behavior reduces light constraints and predation from bats, but may increase potential risk to visual predators such as birds, spiders, and parasitoids. Because the light environment during the day allows for visual aposematism, bright warning coloration is expected (and many diurnal species, both moths and butterflies, are more colorful than nocturnal moths[58]). Non aposematic *Heliconius* butterflies, for example, are slower than their counterparts because they are chemically defended[59], similarly there are many brightly colored, palatable mimics[60].

For species that switched to a nocturnal lifestyle from an ancestrally crepuscular state, light intensity was a limiting factor, and their eyes evolved to become superposition eyes. Superposition eyes effectively act as a large lens that increases the light available to each photoreceptor and these eyes have independently evolved in several nocturnal and crepuscular species[44,61]. The nature of visual pigments makes capturing color information harder as light intensity decreases. Stacking opsins for more sensitive receptors comes at the cost of getting a more broadband signal and a loss of color discrimination[62]. Color vision is slower than monochromatic vision and because the moth visual system also slows down to detect more light at night, seeing color is even more costly at night. The maintenance of trichromacy in nocturnal Lepidoptera is therefore puzzling: perhaps color vision has some critical function in nocturnal Lepidoptera or opsins are maintained for functions independent of color vision.

As a critical function, color can serve as a short-range cue useful for mating and foraging, such as tiger moths that can distinguish conspecifics using color markings that are unrecognizable to birds[63], or the strong innate attraction of flower foraging Lepidoptera to blue[64,65]. It is also possible multiple opsins may be maintained together for other reasons, for example, moths may overcome dim light constraints by pooling from spectrally distinct opsins to get a better representation of various visual stimuli. The most detailed study of opsin distribution across a moth eye shows that there are very different dorsal-ventral patterns in *Manduca* compared to butterflies[66], suggesting that moths may have partitioned color and spatial vision in different regions.

Alternatively, selection could act on each opsin independently, regardless of their utility for color vision. UV and LW opsins models consistently showed differences across models. LW opsin recovery was almost complete, but UV and Blue opsin recovery was patchy. If this trend represents actual losses, though unlikely, it supports the idea that each opsin is maintained independently and LW might be more critical. UV light sensitivity is prevalent in nocturnal animals, even dichromats such as rodents, owls and deep-sea fishes[23,67,68]. UV contrast is a foraging cue for moths, for example nectar guides in many-night blooming flowers[69], and UV light is commonly used as an attractant to moth traps[70–72]. Because short wavelengths increase around twilight[73], UV light is a possible signal for pupil responses, (anecdotally more prevalent across nocturnal moths than butterflies), which could explain the lower selection rates in nocturnal species. LW sensitivity is useful for oviposition behavior in butterflies[74] and their high recovery and low selection rates in nocturnal species could be a signature of their role in oviposition. Butterflies display strong host plant specificity compared to moths[58,75,76], and butterfly LW opsins may have diversified for finer oviposition site discrimination. Irrespective of their role in color vision, LW and UV opsins may

diversify in diurnal species but be held under stabilizing selection in nocturnal species due to integral functions.

In summary, we provide compelling evidence that patterns of opsin diversity and evolution across Lepidoptera are correlated with differences in diel-niche and opsin duplication and rates of evolution. Since light environment can drive gene evolution, there is a high chance that extreme increase in artificial light also plays a similar role. With the current rate of declines in insect populations it is imperative we understand the effects of light environment, especially artificial light on nocturnal animals, and prevent further declines. Future studies on fine-scale expression patterns in closely related diel-transitions can examine opsin expression levels and immunohistochemistry to determine if the duplications are functional. In-vitro expression systems and CRISPR could also confirm that the amino acids identified result in spectral tuning of the photopigment. Examining opsin distribution patterns in eyes across lepidopterans could help to understand if there are sensitivity and color vision trade-offs in different eye regions. This study provides a library of opsin sequences, useful for these sorts of studies, as well as opsin diversity patterns across different species, which can help inform future research.

## Methods

**Lepidoptera opsin gene annotation**. We annotated 162 Lepidoptera transcriptomes obtained from previously published studies (Supplementary Data 1). See supplementary information (Dataset S1, S3, S4) from Kawahara et al.[35] for details on transcriptome assemblies. Transcriptomes were annotated using Phylogenetically Informed Annotation (PIA)[40]. PIA is a bioinformatic pipeline that queries transcriptomes using pre-existing reference gene-sets and places them on a supplied amino acid gene tree. It identifies reading frame and creates a comprehensive gene tree. A modified version of this pipeline was used for faster analysis on a high-performance cluster (https://github.com/xibalbanus/PIA2).

A set of well characterized metazoan visual opsin genes[40] were used as the reference gene set for PIA, and parameters were set to ensure high fidelity of hits while allowing for partial length matches (minimum amino acid sequence length: 30, gene search type: single, gene set: r_opsin, e-value threshold: $e^{-19}$, maximum of blast hits retained: 100) We added opsin sequences from 13 Lepidoptera species obtained from BLAST searches of Lepbase[36], a repository of Lepidoptera genomes and transcriptomes, and Ensembl Metazoa[37]. We used *Manduca sexta* RH4, RH5, RH6, and RH7 opsin sequences as queries for the BLAST search (e-value: 1.0e−10, num_alignments: 250).

The PIA analysis was conservative, it did not misidentify sequences, but it often picked up partial length opsins. If there was only one opsin in that class, we retained it, however for putative duplications we used the local alignment option in Geneious v. 10.0.9 (https://www.geneious.com) to ensure they were not overlapping fragments, in most cases the duplications were easy to identify as duplications. In the cases where they were identical sequences, we retained the longer sequence and did not count that as a duplication. The Lepbase and Ensembl BLAST searches often mischaracterized which opsin family the genes belonged to and the opsin gene tree picked up on these errors (Supplementary Fig. 2).

**Opsin gene tree reconstruction**. The Lepidoptera cDNA sequences were collected and annotated using PIA as putative short wavelength (UV/RH4), medium wavelength (Blue/RH5), long wavelength (LW/RH6), and RH7 (Table S1). However, as a confirmation, we also constructed gene trees for all the sequences. Both trees were midpoint rooted using Archaeopteryx v0.9917beta (https://github.com/cmzmasek/archaeopteryx-js).We used FigTree v1.4.3 (http://tree.bio.ed.ac.uk/software/figtree/) to order and color the nodes and convert the trees into cladograms for easier visualization (Supplementary Figs. 1, 2).

**Gene tree estimation**. MAFFT (v7.294b)[77] was used with default settings to align the sequences and IQ-TREE (multi-core v1.6.12)[78] to build a ML nucleotide gene tree (iqtree-s alignment_name.fasta -st DNA -bb 10000 -nt AUTO -alrt 1000) and an amino acid gene tree (iqtree -s alignment_name.fasta -st AA -bb 10000 -nt AUTO -alrt 1000). IQ-Trees uses ModelFinder[79] to identify the best model and ultrafast bootstrap[80] to calculate bootstrap scores. We used the gene trees to confirm opsin protein annotations taken from the genomes.

**Diel niche assignment**. We assigned diel-niche to compare opsin evolution, but only did so for species which recovered at least one opsin, limiting further analysis to these taxa. Species were classified as "diurnal", "nocturnal", "crepuscular" (species active at dawn or dusk) or "both" (species with some activity during the day and night) (Supplementary Data 1). The diel-niche was assigned using

published literature, natural history databases and in consultation with experts for more obscure species. Diel-niche is often assigned based on whether an insect is attracted to light, and can be a reliable indicator of species that are strictly nocturnal and diurnal[34]. But it often fails for crepuscular species, and species that fly during the night and the day (both). For example, even though *Manduca sexta* is often considered crepuscular, one study has found it is almost entirely nocturnal when compared to *Hyles lineata*, which is active both during the night and day[81]. We used different approaches for diel-niche assignment depending on analyses. For easy tree visualization, we used three diel states, grouping "crepuscular" and "both" into a single category (Fig. 1). For statistical analysis, however, we tried all possible grouping combinations (Table 1). Because some diel-niche assignments are uncertain, we note this ambiguity with a "?" in the dataset. For selection analysis, models only allow two groups, and we therefore categorized species into strictly "nocturnal" and "diurnal" by assigning "crepuscular" and "both" to the "diurnal" category.

**Species-tree reconstruction**. In order to obtain a well-resolved Lepidoptera species tree, we pruned the species tree[82] from Kawahara et al.[35], using Python scripts using the package ETE v3[82], to include only species that had at least one identified opsin. These scripts then modified the pruned tree to show the number of opsins in various Lepidoptera (Fig. 1).

**Opsin duplication**. The python scipy stats package[83] was used to analyze the opsin duplications and their association with diel-niche across Lepidoptera. Custom scripts (Supplementary Information) were used to filter data—including only species for which we recovered at least one opsin—and redo the tests after excluding taxa with uncertainty or varying placement of "crepuscular" and "both" diel-niches as either "nocturnal" or "diurnal" (Table 1). Duplication events resulting from the tree reconciliation analysis were also analyzed (Table 1) but duplications events at deep nodes were ignored for the purpose of the diel-niche associations (Supplementary Data 3) since there is uncertainty about ancestral diel-niches. Chi-squared tests were used to check if the proportion of duplications in each diel-niche was significantly different from a random distribution. Number of species with duplications and number of duplications events were both tested. Since there was no easy analog for number of lineages without duplications, we chose to the most conservative estimate and use the number of diurnal and nocturnal taxa. The Yate's correction for continuity was applied for the duplication event data implemented using a web tool (https://www.socscistatistics.com/tests/chisquare/). The Yate's correction is useful when the event numbers are low in any one category. We also compared the opsin counts for all opsins and for each opsin for extant diurnal and nocturnal taxa used the non-parametric Mann–Whitney U test implemented using scipy.stats package[83] after testing for normality using a D'Agostino and Pearson's test.

**Selection analyses**. The annotated opsin dataset from Lepidoptera was used for estimating rates of selection for the opsins and comparing them across diel-niche. Each opsin family was analyzed separately, but we limited the analysis to taxa which had recovered at least one copy of all three visual opsin sequences (UV, Blue, and LW) to make the analyses comparable between genes and datasets. Because RH7 was only used as a control, we used all taxa for which we recovered RH7. Geneious v. 10.0.9 (https://www.geneious.com) was used to sort and export the sequences. The sequences were manually cleaned in AliView v. 1.18.1[84] and Muscle v. 3.8.31[85] was used for aligning them. Sequences that were too long (<1200 base pairs) were trimmed and sequences shorter than 850 base pairs were removed. Further processing involved manual removal of highly divergent sequences with re-alignment till we obtained no major gaps in the alignment. After a mostly clean alignment was obtained, the sequences were trimmed from either ends to ensure the alignment contained no stop codons or incomplete codons. Final alignments were about 1000 nucleotide bases, roughly 330 amino acids, which falls within the length of characterized opsin sequences[86]. For RH7, the manual method resulted in a very short alignment, less than half the length of the other opsins, so Prank with TranslatorX[87] was used for RH7 even though it had more gaps. IQ-TREE (v. 1.6.12)[78] was used for building the gene trees from the alignments. PAML 4.9a[55] was used to generate various models of codon evolution and estimate site-wise synonymous ($\alpha$) and non-synonymous ($\beta$) rates. The likelihood ratio test was used to determine if a site has a significantly deviant $\beta/\alpha$ (w) from the neutral/null model. We used branch, site, and branch-site models from PAML to test for positive selection and Contrast-FEL[50] and MEME models[51] from HyPhy[52] using the Datamonkey[88] server to verify the branch site models.

Custom python scripts were used to filter the data by number of opsins annotated, compile sequences for the filtered species, and prune the species-tree for the analysis in PAML (Supplementary Information). We tested whether diel-niche ("nocturnal" or "diurnal") has influenced opsin rate evolution for the various selection models. For selection analyses, we categorized the background branches as strictly "nocturnal" and everything else as "diurnal" and as a foreground branch. We ran these analyses under different conditions, varying the number of taxa, choice of gene tree versus species trees and using different alignment methods to test for sensitivity to these parameters (Supplementary Data 4).

**Tree reconciliation**. We used Notung (v-2.9.1.5)[89,90] to perform the tree reconciliation analyses. All trees were pruned using custom python scripts that used the package ETE v3[82]. The pruned species tree and gene trees from the mapping analysis were renamed using python scripts to fit the Notung input format requirements (Supplementary Figs. 3–5). Tree rearrangement minimized the duplication or lost cost to obtain an accurate estimate of duplications and losses in the tree[91].

**Quality control**. We ran two main checks on data quality in order to evaluate potential biases in sampling. First, we used the insect core-ortholog set BUSCO v4.0.6[39] to evaluate overall transcriptome quality, with the percentage of complete orthologs as a proxy. We then reasoned that while some transcriptomes may be high quality, it is possible that the transcriptomes excluded the head of the insect, or that the head may have been very small. In this case, the transcriptome may appear to be high quality (high BUSCO score) but would express few visual genes. To address this case, we used OrthoFinder[92] was used to create a visual gene quality score. Peptide files and a single peptide bait file with known *Manduca sexta* visual genes compiled by a previous study[26] were included in the OrthoFinder run. We removed all '?' characters in the bait file. The clustering algorithm was used to determine orthology to known visual genes. Then, for each sample, we marked presence (1+ gene copies present) or absence (no copies present) of each gene family. The percentage of visual gene families present in each transcriptome (out of a total of 47) serves as a visual gene quality assessment.

**Phylogenetic signal and Phylogenetic ANOVA**. We used the ape v. 5.3[93], phytools[94] and geiger[95] packages in R. Code was adapted from (https://lukejharmon.github.io/ilhabela/instruction/2015/06/02/ContinuousModels/). To test for phylogenetic signal, we used Blomberg's K from the phylosig module in phytools. We tested the phylogenetic ANOVA with the model "quality metric ~diel-niche", with the correlation set to Brownian.

**Ancestral state reconstruction**. We used the geiger[95] package in R with the SYM model and 10,000 repetitions, then used ape v. 5.3[93] to plot on the pruned tree. We generated 10,000 stochastic maps for each tree in SIMMAP[96], which is part of the R package phytools[94]. The methods were adapted from Kawahara et al.[34], which also mapped diel state, without prior information on opsin transition probability. SYM model was used for each opsin class, since we did not want to assume anything about differences in rates for losses versus gains. Stochastic character mapping is a Bayesian approach, supposedly better and more robust than other parsimony or likelihood methods because it allows changes along branches, not just tips, and makes use of data along the nodes to make predictions. It also permits the assessment of uncertainty in character history due to topology and branch lengths[97]. SIMMAP does not allow for missing or unknown data. Therefore, all tips were coded with a discrete, unordered character state, and the taxa with missing traits were pruned from the dataset, causing some discrepancies when comparing reconstructions of opsin number and diel state (Supplementary Fig. 6). We used UV, Blue, LW, total number of opsins, and diel-niche as the discrete characters for the ancestral state reconstruction.

**Protein modeling and site mapping**. We obtained predictions of transmembrane helix prediction for all three opsins using Phobius[98] implemented through Protter[99] using the first sequence from the branch-site alignments (Supplementary Fig. 11). Protter is a web-server based tool useful in annotating a protein sequence. It uses structural prediction tools, such as Phobius[98] to predict transmembrane domains, and allows the user to mark a custom set of amino acids. Only the UV opsin sequence recovered all 7 transmembrane domains, similar to the X-ray crystal structure of known invertebrate opsins and GPCR's, whereas both Blue and LW opsin only recovered six transmembrane domains. UV opsin also had consistently significant differences between diel-niche across the selection analyses and therefore, we used it for the 3D protein structure modeling using an online protein modeling tool, Swiss-model[100]. We chose the squid opsin X-ray structure as a template (2z73.A), because it had the highest identity-score and coverage (GMQE,0.73, identity = 33.6). We included retinal in the model from the squid structure and identified putative retinal binding sites, i.e., amino acids less than 4.0 Å from retinal (Fig. 3A, B). 4.0 Å is the length of weak hydrogen bonds (that longest bonds that opsin usually makes with retinal)[101,102].

**Statistics and reproducibility**. Chi-squared tests were used to assess differences in number of duplication events and total number of duplications for each diel-niche (Table 2) and the duplication events were corrected for Yate's continuity test. Non-parametric Mann–Whitney U tests were used to assess differences in opsin counts (Supplementary Data 1) after testing for normality using a D'Agostino and Pearson's test. ANOVA was used to test if transcriptome quality and visual gene recovery were correlated with diel-niche. Number of runs, support values and input parameters used for the phylogenetic reconstructions, such as the gene trees, ancestral state reconstruction and evolutionary rate estimation are mentioned in the methods and if possible the configuration files and data used at each step for each run are included in Supplementary Data[38]. The protein modeling and associated fits for each model are also included.

**Reporting summary**. Further information on research design is available in the Nature Research Reporting Summary linked to this article.

## Data availability
All datasets, scripts and configuration files that support the findings of this study are available as Supplementary Data[38] on Dryad with the identifiers (https://doi.org/10.5061/dryad.gmsbcc2kr)[38].

## Code availability
Code required to recreate the figures and perform the statistical tests is available on Dryad (https://doi.org/10.5061/dryad.gmsbcc2kr)[38].

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

## Acknowledgements

We thank Jessica Liberles and Joseph Ahrens for guidance with bioinformatic analyses. Heather Bracken-Grissom and Jorge Perez-Moreno for discussions, advice and assistance with bioinformatics pipelines, Ravindran Palavalli-Nettimi, Jesse W. Breinholt for feedback on manuscript drafts. Scott Cinel, Harlan Gough, David Plotkin, Ryan St. Laurent, Caroline Storer, and other members of the Kawahara lab for their assistance throughout the project. John P. Currea, Nick Palmero, and Carlos Ruiz, along with other members of the Theobald lab for comments and feedback on the manuscript. We are grateful the two anonymous reviewers for the comments and suggestions. We acknowledge University of Florida Research Computing for providing computational resources and support that have contributed to the research results reported in this publication (http://researchcomputing.ufl.edu). Funding support for the project was through NSF DEB-1557007 and NSF IOS-1750833. The Florida International University Presidential fellowship supported Y.S.

## Author contributions

Y.S.: conceptualization, methodology, software, visualization, writing–original draft, data curation, formal analysis, project administration. E.E.: conceptualization, methodology, software, data curation, resources, writing–review and editing. S.B.: methodology, validation, writing–review and editing. J.T.: writing–review and editing, visualization, validation, conceptualization. A.K.: conceptualization, methodology, resources, writing–review and editing, visualization, supervision, funding acquisition.

## Competing interests

The authors declare no competing interests.
