## [Peer Review File · Communications Biology]

Reviewers' comments:

Reviewer #1 (Remarks to the Author):

In this paper, Sondhi and colleagues use transcriptome assemblies from a large number of Lepidoptera to examine correlations between light environment and changes in color vision genes (opsins). The authors have assembled an extensive dataset to address this question, with a significant amount of diversity in light environment across it. While this manuscript is of potential interest, I have some significant concerns about the data and methods that need to be addressed.

1. Data quality. The authors rely on a large set of transcriptome assemblies, supplemented by a small number of genome sequences. Transcriptome assemblies are usually pretty incomplete, and the ones analyzed here are no exception (e.g. lines 98-126). I can think of at least three possible problems that arise from this amount of missing data.

A. The most obvious issue is on inference of loss of opsins: the authors attempt to distinguish between losses of genes due to incomplete sequencing and losses of genes due to actual biological loss by using ancestral state reconstruction. If losses are randomly distributed across the tree, then ancestral nodes should tend towards a uniform probability of loss, under the assumption that sampling effort (and probability of loss) is also uniform across the tree. If some clades are sequenced more deeply and others less deeply then of course there could be a strong phylogenetic signal of loss associated with sequencing depth, which would not necessarily indicate true biological losses in those species. The authors as far as I see do not do any tests to show there ancestral state reconstruction method is expected to work, instead of picking up clade-specific differences in assembly quality. Analyzing control genes, or more careful mapping of transcriptome properties (BUSCO scores, sequencing depth, number of genes) to the phylogeny may help resolve this.

B. Another concern is regarding the possible rate of failure to detect real duplication events. The authors do not explicitly discuss this possibility, but it seems at least feasible that in some lower-coverage transcriptomes, one copy of a particular opsin might be detected while a second, biologically real copy could be missed. This would be of largest concern if transcriptome quality is phylogenetically correlated, as then real duplicates could be missed in many related species. This could also be an issue if, for some reason, average transcriptome quality is higher in diurnal than nocturnal species.

C. A final concern involves the molecular evolution analysis with PAML. Examining the raw data files, there are huge differences between the final trimmed and cleaned versions of the alignments used in PAML, and the original alignments in e.g. Dataset 2. It seems this is mostly the result of extensive manual processing in Geneious, which is not well described. It is important to know by what criteria sequences were considered "too short" or "too long" and if they were too long, how the ends were trimmed. Alternatively, an automated method to filter out non-homologous bases (e.g. PREQUAL) may be more robust and reproducible than manual trimming. Transcriptome assemblies can easily produce models that include non-homologous bases (e.g. incorrectly included introns).

Ultimately I think these data quality issues are addressable, but at the moment point to concerns about the robustness of the author's conclusions.

2. Methods.

A. The authors rely on ancestral state reconstruction to identify changes in gene copy numbers at ancestral nodes. However, because they have both a resolved species tree and gene trees for individual genes, it should be possible to use gene tree / species tree reconciliation methods to more precisely identify lineages on which duplications and losses have occurred. This would allow a

more precise correlation between transitions in phenotype and gene duplications. It is not clear, for example, if Table 1 reflects the number of duplication events in diurnal vs nocturnal species, or the number of species with duplications. If the latter, this overstates support for a correlation. This is a critical issue given its centrality to the manuscript's conclusions.

B. The authors primarily rely on PAML branch-site models to infer evidence for positive selection in diurnal vs nocturnal species. These tests are known to suffer from high false positive rates under certain conditions (including multinucleotide mutations and incomplete lineage sorting or other sources of gene tree incongruence). While the branch tests, especially for species tree analysis, should be reasonably robust to these issues, the branch-site tests are likely more susceptible. The site models show no evidence at all for positive selection. Typically selection does not need to be occurring in all or even most species to detect it with site models, so given that diurnal and nocturnal species are relatively evenly split in the PAML dataset, the lack of signal in site models is a bit concerning and suggests the possibility that the branch-site results are artifactual. This is not a perfect test, of course, but the potential for false positives in the branch-site test needs to be addressed.

3. While the points above are the main concerns, there are some small terminology and other minor issues that should be addressed. Listed by line number below:

70-71: Citation needed for "both of which can bias results"

100-104: As discussed above, need to clarify if this is number of duplication events or something else

156-159: As discussed above, more detail needed in trimming protocol

166: Do authors mean relaxed selection? The manuscript does not use consistent terminology to refer to lineages with increased dN/dS (see also line 178, other places). May be simplest to just say higher dN/dS.

199-201: More sites under selection is not evidence that species tree models are more accurate; could reflect false positives

Reviewer #2 (Remarks to the Author):

In this study, the authors hypothesize that the evolution of opsins, including duplications, losses, and diversification, is correlated with diel niche in Lepidoptera. To test this, the authors annotated opsin proteins in 200 existing insect transcriptomes (including 176 Lepidoptera) and compared opsin diversity across the species tree. They found that in diurnal species, opsin expansions occurred more often and evolved more quickly than in nocturnal species. Key amino acid residues also were identified as likely spectral tuning sites. Overall, the results are important for understanding the relationship between light environment and opsin evolution.

The authors have assembled a large, albeit, complex dataset and performed relatively thorough analyses. However, the study still has a few major issues that need addressed prior to publication.

Major revisions:

1. First, I encourage the authors to correct the numerous typographical errors present throughout. A simple spell-check will suffice. Along these lines, it is encouraged to not leave uncorrected "Track changes" in the manuscript from the co-authors (lines 674-679) and properly follow a consistent citation format (e.g., 358).

2. Although extensive data are available to allow the authors to explore opsin diversification in Lepidoptera across diel-switches, the data collected across 50 insect species seem unfortunately irrelevant. The authors discuss that this serves as a 'positive control for the annotation pipeline' (lines 91-93), but don't describe this in enough detail. How exactly does this serve as a positive

control? This is the Results section, so quantitative values would be useful - for example, what proportion of known opsins encoded in the genome (of those species with a genome sequence) were recovered with the transcriptomes? This is extremely important (see major comment below), as it relates to the potential for false negatives. Currently, the authors state that only 28/50 insect transcriptomes recovered an opsin, which seems like there are a large number of false negatives (missing opsins). Otherwise, the insect transcriptomes contain "too few nocturnal taxa to statistically compare opsin evolution between diel-niches"... so why were these data included? I suggest the authors focus on only Lepidoptera, as these are the only data relevant to answering the hypothesis regarding diel-switches (see statement of limiting discussion to "a dataset of Lepidoptera with sufficient diel variation and better phylogenetic coverage to address this question" lines 94-96).

3. The primary weakness of this study is (to quote the authors lines 108-110): "Opsin losses are notoriously difficult to confirm, since inferred losses could be due to poor opsin recovery or low-quality transcriptomes". The next few lines, 110-113 (also 240-248) describe an altogether weak justification for including this analysis, especially considering the likely large number of false negatives. The authors keep alluding to this issue:

- lines 117-119 (UV opsin): "We did not find multiple lineages with a high probability of loss, but deeper nodes had a 50% chance of loss, likely indicating incomplete data
- lines 126-127 (Blue opsin): for all but four lineages, "the chance of a loss was about 50%, likely indicating issues in opsin recovery and annotation."
- lines 149-150 (RH7): "sample collection affected the expression of opsins and might explain the lower recovery rate across our dataset"

As mentioned above, some kind of 'positive control' is warranted, especially in Lepidoptera. At minimum, I suggest the authors compare their transcriptome results with those few species having a high quality reference genome, to determine the recovery rate of the 'true' number of opsins in a species. At this point, without such data, and considering the huge variability in transcriptomes (tissue source, library prep and sequencing methods, assembly methods [e.g., trinity produces numerous chimeric/isoform contigs], QC criteria, etc) assessing 'absence' is highly questionable. A potential response to this would be rather to focus on the evolutionary rates (PAML) analysis on the opsins recovered, as those data are more reliable (aren't based on absence) and nicely support the hypothesis.

4. Lines 152-205: The authors need to include proper description of their results and the output of statistical tests. In other words, this entire section has very few quantitative values reported to convince the reader of their findings. The branch/branch site tests rely on a likelihood ratio test to a null model, but the key test results are never presented in the main text (chisquare value, p-value, sample, size, etc). The authors should refer to general APA guidelines for reporting test results. Furthermore, was proper adjustment taken to correct for multiple testing?

5. With regards to the protein modeling findings (lines 208-220). The authors report that sites under positive selection are close to the retinal binding regions. However, this needs to be put in more context. For example, what is the average distance of all amino acids from a retinal binding region? In other words, it isn't clear how often you would expect an amino acid at random to be within the minimum distance to retinal, and if the sites under selection are closer than expected by chance.

6. Lines 256-258: "The ancestors of both diurnal and nocturnal lepidopterans were likely crepuscular with limited trichromatic vision, so what is likely to have driven this selection?" Here the authors cite their own work (lead PI/senior author) despite that this work clearly contradicts their statement. From citation #29: "our analysis inferred that diurnality is the likely ancestral condition in the order". The authors need to clarify this issue.

- Minor Revisions

Lines 30-37: This introductory paragraph is awkward and reads like a second abstract (no citations

are provided as well). I recommend removing this paragraph, or at minimum extensive revisions.

Lines 38-39: what about other aspects of vision, contrast, edge detection, brightness, etc?

Lines 42-43: I recommend including a more relevant introduction to non-ocular vision here, for example in dermal photoreception, circadian rhythm entrainment, etc (especially insect examples) - bacteria and fungal spores seem not a useful comparison when better example exist.

Lines 44-51, 67-80: Unfortunately, for an article about opsins, the authors need to improve the introductory material with regards to opsins. What are the different opsins described (especially in Lepidoptera)? This is important because the entire results and discussion centers around the various opsin types (LWS, RH7, SWS, UV, etc) which aren't introduced to the reader until the results section. Furthermore, referring all opsins as 'rhodopsin' (line 45) is either factually incorrect or a gross oversimplification.

Lines 53-54: Opsins have been described in numerous non-visual functions (although often light-sensitive in some way. e.g, skin pigmentation, dermal photoreception, etc).

Lines 58-59: Is there evidence to support this claim or is it speculation?

Lines 62-63: My only concern here is that abundant light will not cause more duplications, but rather prevent the loss of duplicated opsins. This is a subtle, but important distinction.

Lines 89-90: What does it mean to 'reconfirm the annotations using the nucleotide and amino acid trees? Can you please provide a bit more detail?

Line 102: These opsins and their abbreviations need to be described ahead of time and the nomenclature used consistently throughout the manuscript.

Lines 159-162: It is more than sufficient to include good alignments with accurately introduced gaps. These alignments can be cleaned with programs like Gblocks or TrimAL, and CODEML will actually ignore these gaps (the supplementary CODEML files, codeml.ctl, already have set "cleandata = 1", which removes alignment gaps).

Lines 202-204: "Since species tree models predicted more sites in the helical regions—which can affect and tune the opsin—we infer that the species tree models are more accurate." I cannot understand the rationale for this. Please clearly explain this, and/or perhaps it is better suited to the discussion section.

Lines 253-255: Does the shift to diurnality drive opsin diversification, or does a diverse repertoire of opsins (post duplication), allow species to invade new light environments? The authors should elaborate on this a bit more.

Lines 262-265: This statement doesn't appear relevant and it is suggested to be removed.

Lines 275-278: Could the authors elaborate on this statement a bit more?

Lines 333-334: This seems inadequate. What does 'manual curation' entail? Please clarify. How exactly were putative duplications determined, aligned, and assessed for not being fragments of the same gene. This is a notorious problem in transcriptome assemblers.

Response to Reviewers

Title: *Light environment drives evolution of color vision genes in butterflies and moths*

ms#: COMMSBIO-20-0512-T

Authors: Yash Sondhi, Emily A. Elis, Seth M. Bybee, Jamie C. Theobald, Akito Y. Kawahara

General response to reviewers:

We appreciate the valuable insight from the reviewers and their helpful suggestions for improving the manuscript. We agree with the reviewers' concerns, and to fully address them we have solicited the help of Dr. Seth Bybee, who has expertise with the analyses suggested, and now include him as a co-author. To address the reviewers' concerns, we ran quality control metrics, mapped them onto the phylogeny, and checked for correlations with diel-niche. However, we find no link between transcriptome quality and either phylogeny or diel-niche. To verify the results of our selection models, we repeated the analyses with newer algorithms and found similar, but statistically significant results. Reviewer #2 raised an important point about species with duplications vs. duplication events. Running tree reconciliation analyses resulted in the same trend with duplication events that we had previously found when comparing species with duplications. We gained additional insight regarding which lineages they occurred in and how old they were, ultimately strengthening our findings. As suggested by reviewer #2, we removed the section about insect opsins and all references to those analyses, which tightens our main message. Below, we outline the specific changes we made to the manuscript. Where required, we justify or explain our reasoning. We hope that the manuscript is acceptable now that nearly all of the reviewers' major and minor concerns have been incorporated. We thank the reviewers for ideas that substantially improved the manuscript.

Reviewer #1 (Remarks to the Author):

Comments: *In this paper, Sondhi and colleagues use transcriptome assemblies from a large number of Lepidoptera to examine correlations between light environment and changes in color vision genes (opsins). The authors have assembled an extensive dataset to address this question, with a significant amount of diversity in light environment across it. While this manuscript is of potential interest, I have some significant concerns about the data and methods that need to be addressed.*

1. Data quality. The authors rely on a large set of transcriptome assemblies, supplemented by a small number of genome sequences. Transcriptome assemblies are usually pretty incomplete, and the ones analyzed here are no exception (e.g. lines 98-126). I can think of at least three possible problems that arise from this amount of missing data.

A. The most obvious issue is on inference of loss of opsins: the authors attempt to distinguish between losses of genes due to incomplete sequencing and losses of genes due to actual biological loss by using ancestral state reconstruction. If losses are randomly distributed across the tree, then ancestral nodes should tend towards a uniform probability of loss, under the assumption that sampling effort (and probability of loss) is also uniform across the tree. If some clades are sequenced more deeply and others less deeply then of course there could be a strong phylogenetic signal of loss associated with sequencing depth, which would not necessarily indicate true biological losses in those species. The authors as far as I see do not do any tests to show their ancestral state reconstruction method is expected to work, instead of picking up clade-specific differences in assembly quality. Analyzing control genes, or more careful mapping of transcriptome properties (BUSCO scores, sequencing depth, number of genes) to the phylogeny may help resolve this.

Response: We thank the reviewer for highlighting this important point regarding gene loss and the suggestion to investigate quality. We agree that inferring loss is problematic, and additionally, obfuscates our strong findings. Instead of interpreting potential losses, we have further focused our messaging towards the interesting expansion and selection of opsins with transitions to diurnality. However, as the reviewer points out in (B), our interpretation of opsin duplication could also be impacted by transcriptome quality metrics as well. As such, we analysed two quality control metrics for the transcriptome dataset, BUSCO scores and percentage recovery of visual genes and mapped it to the phylogeny (see lines 398-409 for details about quality control methods). We tested if these quality metrics had a phylogenetic signal and found they had no significant phylogenetic signal ($K=0.3$), $K>1$ is considered a strong phylogenetic signal. We ran a phylogenetic ANOVA and did not find evidence for a correlation of diel-habitat with quality ($p\text{-value}=0.8$). (see lines 410-414 for details about comparative phylogenetic methods). These additional quality control metrics bolster our confidence in our findings and show that expansions in opsin diversity are not due to a biased representation of quality across our dataset.

Comment: *B. Another concern is regarding the possible rate of failure to detect real duplication events. The authors do not explicitly discuss this possibility, but it seems at least feasible that in some lower-coverage transcriptomes, one copy of a particular opsin might be detected while a second, biologically real copy could be missed. This would be of largest concern if transcriptome quality is phylogenetically correlated, as then real duplicates could be missed in many related species. This could also be an issue if, for some reason, average transcriptome quality is higher in diurnal than nocturnal species.*

Response: As we noted in our response to point (A), we found no correlation between quality metrics and phylogeny. Further, we tested whether quality metrics were correlated with diurnality or nocturnality, as this may also produce the strong pattern we document. Importantly, we found no correlation of BUSCO scores and visual gene score results with diel-habitat after accounting for phylogeny. We ran a phylogenetic ANOVA and did not find evidence for a correlation of diel-habitat with quality ($p\text{-value}=0.8$). (see line numbers 98-104, 410-414 for results and detail about methodology).

Comment: *C. A final concern involves the molecular evolution analysis with PAML. Examining the raw data files, there are huge differences between the final trimmed and cleaned versions of the alignments used in PAML, and the original alignments in e.g. Dataset 2. It seems this is mostly the result of extensive manual processing in Geneious, which is not well described. It is important to know by what criteria sequences were considered "too short" or "too long" and if they were too long, how the ends were trimmed. Alternatively, an automated method to filter out non-homologous bases (e.g. PREQUAL) may be more robust and reproducible than manual trimming. Transcriptome assemblies can easily produce models that include non-homologous bases (e.g. incorrectly included introns).*

Ultimately I think these data quality issues are addressable, but at the moment point to concerns about the robustness of the author's conclusions.

Response: We added significantly more details about trimming to manuscript (see lines 369-380). We relied on manual trimming with the goal being to get a clean alignment region that minimised gaps and had a high confidence that the bases were homologous. The general approach to this was to err on the side of caution and only retain sequences that we approached the average length and sequence similarity of the other opsins in the dataset, we removed highly divergent sequences or

sequences that were very short (<850 nts.) and trimmed the sequences that were very long to try and get about 1000 bp which translates to 330 sites, matching the sequence length of well-known opsins in Lepidoptera. While we agree that there might be more automated methods to do this, we were very conservative and discarded any sequences that reduced the overall quality of the alignment significantly. We are confident that our final alignment is clean since it gives consistent selection coefficients using a number of different programs and models.

Comment: 2. *Methods.*

A. The authors rely on ancestral state reconstruction to identify changes in gene copy numbers at ancestral nodes. However, because they have both a resolved species tree and gene trees for individual genes, it should be possible to use gene tree / species tree reconciliation methods to more precisely identify lineages on which duplications and losses have occurred. This would allow a more precise correlation between transitions in phenotype and gene duplications. It is not clear, for example, if Table 1 reflects the number of duplication events in diurnal vs nocturnal species, or the number of species with duplications. If the latter, this overstates support for a correlation. This is a critical issue given its centrality to the manuscript's conclusions.

Response: Species tree gene tree reconciliation methods using Notung, as the reviewer suggested, helped resolve which nodes the duplications occurred. The analysis mentioned in the paper is the number of species with duplications. We ran UV, Blue and LW gene and species trees with Notung and found the number of overall duplication events in nocturnal and diurnal species was altered only slightly. 14 (74%) duplication events were in diurnal lineages and 5 (26%) duplication events were in nocturnal lineages, of which 3 of those were from the migratory *Spodoptera* lineage, which is not always light limited.

We have included a table (Table S3) with the exact number of duplication events and included the Notung analysis in the supplementary files along with redoing the statistics on them. Since the numbers were almost unchanged from some of the estimates run with number of species with duplications the analysis run earlier, we only include the estimates of the chi-square tests on the most conservative diurnal sample numbers for the chi-square test even though since we are now considering lineages, that number should ideally be higher (42 D and 46 N) in the manuscript ($\chi^2 = 6.0259$ p-value = 0.014, significant at $p < .05$).

Since the numbers are unchanged and we are confident the original interpretation is valid, we mention both duplication events and number of species with duplications in the manuscript.

Comment: *B. The authors primarily rely on PAML branch-site models to infer evidence for positive selection in diurnal vs nocturnal species. These tests are known to suffer from high false positive rates under certain conditions (including multi-nucleotide mutations and incomplete lineage sorting or other sources of gene tree incongruence). While the branch tests, especially for species tree analysis, should be reasonably robust to these issues, the branch-site tests are likely more susceptible. The site models show no evidence at all for positive selection. Typically selection does not need to be occurring in all or even most species to detect it with site models, so given that diurnal and nocturnal species are relatively evenly split in the PAML dataset, the lack of signal in site models is a bit concerning and suggests the possibility that the branch-site results are artifactual. This is not a perfect test, of course, but the potential for false positives in the branch-site test needs to be addressed.*

Response: The PAML branch site is thought to miss instances of episodic positive selection. We implemented newer methods on HyPhy servers such as Contrast-FEL and MEME that are more sensitive to other forms of positive selection and they identified several sites under positive

selection. Further, BEB models in PAML predicted different sites under positive selection, but the Likelihood Ratio test was probably too conservative to pick them especially in the backdrop of the strong pervasive purifying selection acting on the opsins. The sites predicted by the newer methods had some overlap for UV and LW opsins where we saw the strongest signal for selection from the branch models and the amino acids with the lowest P-value scores showed correspondingly low p-values and false discovery scores from the other methods as well. The sites and are included in supplementary tables (Table S4). We think this addresses some of the concerns for false positives in the branch-site models, and we also mention the limitations of these methods in the discussion. (lines 231-240)

Comments: 3. *While the points above are the main concerns, there are some small terminology and other minor issues that should be addressed. Listed by line number below:*

Comments: 70-71: *Citation needed for "both of which can bias results"*

Response: Citation added and the text has been modified to clarify why the bias occurs.

Comment: 100-104: *As discussed above, need to clarify if this is number of duplication events or something else.*

Response: Added and modified the language.

Comment: 156-159: *As discussed above, more detail needed in trimming protocol.*

Response: Added more detail to the trimming protocol

Comment: 166: *Do authors mean relaxed selection? The manuscript does not use consistent terminology to refer to lineages with increased dN/dS (see also line 178, other places). May be simplest to just say higher dN/dS.*

Response: Changes made and checked for consistency

Comment: 199-201: *More sites under selection is not evidence that species tree models are more accurate; could reflect false positives.*

Response: Removed statement and added a more appropriate justification for using the species trees, the earlier argument was unjustified and circular.

Reviewer #2 (Remarks to the Author):

In this study, the authors hypothesize that the evolution of opsins, including duplications, losses, and diversification, is correlated with diel niche in Lepidopterans. To test this, the authors annotated opsin proteins in 200 existing insect transcriptomes (including 176 Lepidopterans) and compared opsin diversity across the species tree. They found that in diurnal species, opsin expansions occurred more often and evolved more quickly than in nocturnal species. Key amino acid residues also were identified as likely spectral tuning sites. Overall, the results are important for understanding the relationship between light environment and opsin evolution. The authors have assembled a large, albeit, complex dataset and performed relatively thorough analyses. However, the study still has a few major issues that need addressed prior to publication.

Comments: *Major revisions:*

1. First, I encourage the authors to correct the numerous typographical errors present throughout. A simple spell-check will suffice. Along these lines, it is encouraged to not leave uncorrected "Track

changes" in the manuscript from the co-authors (lines 674-679) and properly follow a consistent citation format (e.g., 358).

Response: Thank you for this suggestion. We used spell-checking again and proof-read the manuscript for errors multiple times.

Comment: 2. *Although extensive data are available to allow the authors to explore opsin diversification in Lepidoptera across diel-switches, the data collected across 50 insect species seem unfortunately irrelevant. The authors discuss that this serves as a 'positive control for the annotation pipeline' (lines 91-93), but don't describe this in enough detail. How exactly does this serve as a positive control? This is the Results section, so quantitative values would be useful - for example, what proportion of known opsins encoded in the genome (of those species with a genome sequence) were recovered with the transcriptomes? This is extremely important (see major comment below), as it relates to the potential for false negatives. Currently, the authors state that only 28/50 insect transcriptomes recovered an opsin, which seems like there are a large number of false negatives (missing opsins). Otherwise, the insect transcriptomes contain "too few nocturnal taxa to statistically compare opsin evolution between diel-niches"... so why were these data included? I suggest the authors focus on only Lepidoptera, as these are the only data relevant to answering the hypothesis regarding diel-switches (see statement of limiting discussion to "a dataset of Lepidoptera with sufficient diel variation and better phylogenetic coverage to address this question" lines 94-96).*

Response:

We agree with reviewer #2 and adjusted the text to only focus on the Lepidoptera and have removed insect dataset from the manuscript.

Comment: 3. *The primary weakness of this study is (to quote the authors lines 108-110): "Opsin losses are notoriously difficult to confirm, since inferred losses could be due to poor opsin recovery or low-quality transcriptomes". The next few lines, 110-113 (also 240-248) describe an altogether weak justification for including this analysis, especially considering the likely large number of false negatives. The authors keep alluding to this issue:*

- lines 117-119 (UV opsin): *"We did not find multiple lineages with a high probability of loss, but deeper nodes had a 50% chance of loss, likely indicating incomplete data*

- lines 126-127 (Blue opsin): *for all but four lineages, "the chance of a loss was about 50%, likely indicating issues in opsin recovery and annotation."*

- lines 149-150 (RH7): *"sample collection affected the expression of opsins and might explain the lower recovery rate across our dataset"*

-As mentioned above, some kind of 'positive control' is warranted, especially in Lepidoptera. At minimum, I suggest the authors compare their transcriptome results with those few species having a high-quality reference genome, to determine the recovery rate of the 'true' number of opsins in a species. At this point, without such data, and considering the huge variability in transcriptomes (tissue source, library prep and sequencing methods, assembly methods [e.g., trinity produces numerous chimeric/isoform contigs], QC criteria, etc) assessing 'absence' is highly questionable. A potential response to this would be rather to focus on the evolutionary rates (PAML) analysis on the opsins recovered, as those data are more reliable (aren't based on absence) and nicely support the hypothesis.

Response: We agree with the reviewer's comment and will not assess 'absence'. Since we do not find evidence that diel-niche and phylogeny are correlated with transcriptome quality, we think the

duplications linked to diel-habitat and the selection analyses are robust and are the main conclusion of the manuscript. We mention the Tree reconciliation and ASR analysis and add the files but refrain from making any claims about losses or what they might mean and expand on the limitations in the discussion.

New Text: Results

"We performed an ancestral state reconstruction (ASR) of diel-niche, each opsin and total number of opsins and qualitatively, diurnal lineages and nodes have ancestrally higher opsin numbers (Fig. S3A). Opsin losses are notoriously difficult to confirm³¹ and while both tree reconciliation and ASR results are included (Fig. S3 A-D, S5 A-C, Supplementary Information), we refrain from interpreting the losses."

New Text: Discussion

"We are not confident that the losses we identify are true losses and refrain from making claims about trends for losses, but include both the ancestral state reconstruction and tree reconciliation results (Supplementary Information)."

Comment: 4. *Lines 152-205: The authors need to include proper description of their results and the output of statistical tests. In other words, this entire section has very few quantitative values reported to convince the reader of their findings. The branch/branch site tests rely on a likelihood ratio test to a null model, but the key test results are never presented in the main text (chisquare value, p-value, sample, size, etc). The authors should refer to general APA guidelines for reporting test results. Furthermore, was proper adjustment taken to correct for multiple testing?*

Response: Included main results from the supplemental Tables and main Tables in the main text and newer analysis of selection was included where corrections for multiple testing have been included.

Comment: 5. *With regards to the protein modelling findings (lines 208-220). The authors report that sites under positive selection are close to the retinal binding regions. However, this needs to be put in more context. For example, what is the average distance of all amino acids from a retinal binding region? In other words, it isn't clear how often you would expect an amino acid at random to be within the minimum distance to retinal, and if the sites under selection are closer than expected by chance.*

Response: We qualify these results with the bio-physical cut-off distances of various bond lengths, we use the upper bound, and the weakest hydrogen bond length is 4.0 Å. We then examine if the retinal binding sites interact with the amino acids under positive selection, we use 2.0 Å as the cut-off. This reduces the number of amino acids that are likely to be tuning sites, but makes a stronger case for the ones that are in this range. However, this does not discount the possibility that others positively selected sites interact with retinal indirectly, we just do not have the data to make the claim for or against it.

Comment: 6. *Lines 256-258: "The ancestors of both diurnal and nocturnal lepidopterans were likely crepuscular with limited trichromatic vision, so what is likely to have driven this selection?" Here the authors cite their own work (lead PI/senior author) despite that this work clearly contradicts their statement. From citation #29: "our analysis inferred that diurnality is the likely ancestral condition in the order". The authors need to clarify this issue.*

Response: Issue clarified, and is consistent

Comments: - *Minor Revisions*

Lines 30-37: This introductory paragraph is awkward and reads like a second abstract (no citations are provided as well). I recommend removing this paragraph, or at minimum extensive revisions.

Response: Thank you for this suggestion. This paragraph has been removed.

Comment: *Lines 38-39: what about other aspects of vision, contrast, edge detection, brightness, etc?:*

Response: Incorporated feedback and text changed.

Comment: *Lines 42-43: I recommend including a more relevant introduction to non-ocular vision here, for example in dermal photoreception, circadian rhythm entrainment, etc (especially insect examples) - bacteria and fungal spores seem not a useful comparison when better example exist.*

Response: Incorporated feedback and changed the introductory paragraph to be more relevant.

Comment: *Lines 44-51, 67-80: Unfortunately, for an article about opsins, the authors need to improve the introductory material with regards to opsins. What are the different opsins described (especially in Lepidoptera)? This is important because the entire results and discussion centers around the various opsin types (LWS, RH7, SWS, UV, etc) which aren't introduced to the reader until the results section. Furthermore, referring all opsins as 'rhodopsin' (line 45) is either factually incorrect or a gross oversimplification.*

Response: Agree and added paragraph about opsins in introductory paragraph and a second paragraph describing the various types of opsins.

Comment: *Lines 53-54: Opsins have been described in numerous non-visual functions (although often light-sensitive in some way. e.g, skin pigmentation, dermal photoreception, etc).*

Response: We agree and incorporated these suggested changes to the introductory paragraph.

Comment: *Lines 58-59: Is there evidence to support this claim or is it speculation?*

Response: References of specific examples where this has occurred are mentioned in the same example.

Comment: *Lines 62-63: My only concern here is that abundant light will not cause more duplications, but rather prevent the loss of duplicated opsins. This is a subtle, but important distinction.*

Response: Agree and incorporated changes.

Comment: *Lines 89-90: What does it mean to 'reconfirm the annotations using the nucleotide and amino acid trees? Can you please provide a bit more detail?*

Response: Agreed and incorporated changes, but elaborated this in the methods.

Comment: *Line 102: These opsins and their abbreviations need to be described ahead of time and the nomenclature used consistently throughout the manuscript.*

Response: Agreed and incorporated changes.

Comment: *Lines 159-162: It is more than sufficient to include good alignments with accurately introduced gaps. These alignments can be cleaned with programs like Gblocks or TrimAL, and CODEML will actually ignore these gaps (the supplementary CODEML files, codeml.ctl, already have set "cleandata = 1", which removes alignment gaps).*

Response: We agree, and have incorporated appropriate changes to the text. Partially and added more details on methods and included reference to support claim.

Comment: *Lines 202-204: "Since species tree models predicted more sites in the helical regions—which can affect and tune the opsin—we infer that the species tree models are more accurate." I cannot understand the rationale for this. Please clearly explain this, and/or perhaps it is better suited to the discussion section.*

Response: We agree and have incorporated suggestions as well as provided a better justification for the use of species tree, as well as elaborate our rationale in the discussion.

Comment: *Lines 253-255: Does the shift to diurnality drive opsin diversification, or does a diverse repertoire of opsins (post duplication), allow species to invade new light environments? The authors should elaborate on this a bit more.*

Response: Agree and incorporated suggestions

Comment: *Lines 262-265: This statement doesn't appear relevant and it is suggested to be removed.*

Response: Removed requested lines

Comment: *Lines 275-278: Could the authors elaborate on this statement a bit more?*

Response: Elaborated statement in the context of above arguments.

Comment: *Lines 333-334: This seems inadequate. What does 'manual curation' entail? Please clarify. How exactly were putative duplications determined, aligned, and assessed for not being fragments of the same gene. This is a notorious problem in transcriptome assemblers.*

Response: Details added to the methods section.

New Text: *Methods*

Manual Curation

"Geneious v. 10.0.9 (<https://www.geneious.com>) was used to sort and export the sequences. The sequences were manually cleaned in AliView v. 1.18.1⁸⁸ and Muscle v. 3.8.31⁸⁹ was used for aligning them. Sequences that were too long (< 1200 base pairs) were trimmed and sequences shorter than 850 base pairs were removed. Further processing involved manual removal of highly divergent sequences with re-alignment till we obtained no major gaps in the alignment. After a mostly clean alignment was obtained, the sequences were trimmed from either ends to ensure the alignment contained no stop codons or incomplete codons. Final alignments were about 1000 nucleotide bases, roughly 330 amino acids, which falls within the length of characterized opsin sequences⁹⁰. For RH7, the manual method resulted in a very short alignment, less than half the length of the other

opsins, so Prank with TranslatorX⁹¹ was used for RH7 even though it had more gaps. IQ-TREE (v. 1.6.12)⁸² was used for building the gene trees from the alignments.

Assessing false duplications

“The PIA analysis was conservative, it did not misidentify sequences, but it often picked up partial length opsins. If there was only one opsin in that class, we retained it, however for putative duplications, we used the local alignment option in Geneious v. 10.0.9 (<https://www.geneious.com>) to ensure they were not overlapping fragments, in most cases the duplications were easy to identify as duplications. In the cases where they were identical sequences, we retained the longer sequence and did not count that as a duplication. The Lepbase and Ensembl BLAST searches often mischaracterized which opsin family the genes belonged to and the opsin gene tree picked up on these errors (Fig S2).”

Reviewers' comments:

Reviewer #1 (Remarks to the Author):

The authors have done an admirably thorough job responding to the previous reviews, and I think have addressed all my concerns. The revised manuscript reads well and focuses appropriately on the strongest conclusions from the analysis. I do not think any further revisions are necessary.

Reviewer #2 (Remarks to the Author):

In this revised manuscript, the authors did an excellent job addressing my previous concerns. I am very impressed at the amount of revisions made in such a short time. The main analyses are now much clearer by removing the section on 50 insect transcriptomes and questionable analysis of opsin losses. Furthermore, the assessment of BUSCO/vision scores across the phylogeny helps support that variation in assembly quality may not necessarily be related to the evolutionary signals detected. Admittedly, I am not yet fully convinced that such variation in transcriptome quality, tissue, methods, etc can be robustly compared from a copy-number standpoint, but the authors have done a sufficient job attempting to account for this in the best possible manner and have appropriately acknowledged its limitations. Overall, I just have one moderate and several minor remarks below that should be taken into consideration. Otherwise I support the publication of this study.

lines 105-106: the authors mention that "qualitatively" it appears that ancestral diurnal lineages have more opsins. Is there a way to quantitatively compare this? Would it be possible to do something like a Mann-whitney U test to compare opsin counts on diurnal vs nocturnal ancestral branches? The signal appears to be there from the figure, but quantitative assessments are preferred.

lines 148,150,154: p-values are not a measure of effect size, thus phrases like "highly significant" and "moderately significant" are not appropriate. These should simply be phrased as significant or not, then effect sizes compared.

lines 158-169: This is the only section that I find troublesome. Upon reading, it sounds as if the authors attempted multiple evolutionary tests until they found one that supported their conclusions. For example the PAML models found "no significant signature of positive selection" (line 159) and the "LRT tests were not significant" (lines 161-2). Despite describing the LRT tests as "fairly conservative" (lines 162-3), subsequent, "more sensitive" (line 163,168) tests were applied in HYPHY. These new tests then report results consistent with the hypothesis. My point here is that if the HYPHY models are "tailored to detect specific kinds of evolutionary scenarios" (line 168-9), then please provide some kind of support/justification or description of why the reader should trust these results more than those from PAML, considering their conclusions are conflicting.

lines 177-187: this entire section describes methods and not results. I recommend placing in the 'Methods' section.

lines 178-179: Should the reader be concerned that only 6 transmembrane domains were identified in most opsins other than the UV opsins? Having 7 transmembrane domains is a signature of these GPCRs.

lines 197-204: this section does not include any results but rather interpretation, I recommend moving to the "Discussion" section.

lines 240-241: Again, see my comment above regarding the justification of using the Hyphy

results rather than PAML results.

Response to Reviewers

Title: *Light environment drives evolution of color vision genes in butterflies and moths*

ms#: COMMSBIO-20-0512-T

Authors: Yash Sondhi, Emily A. Elis, Seth M. Bybee, Jamie C. Theobald, Akito Y. Kawahara

General response to reviewers:

Reviewer #2 (Remarks to the Author):

Moderate revisions

Comments: lines 105-106: *the authors mention that "qualitatively" it appears that ancestral diurnal lineages have more opsins. Is there a way to quantitatively compare this? Would it be possible to do something like a Mann-Whitney U test to compare opsin counts on diurnal vs nocturnal ancestral branches? The signal appears to be there from the figure, but quantitative assessments are preferred.*

Response:

It is difficult to quantitatively test if there are more opsins in ancestral diurnal lineages because of the range of values present at each node. It is theoretically possible to measure this range but the implementation in R is not well developed since most implementations are designed for comparing extant taxa. We reached out to the several researchers including the author of the R package (PhyTools) that would potentially implement this analysis but have not been able to find an implementation. Even if we could implement it there are several issues. First, the definition of what is diurnal from the ancestral reconstruction gets more uncertain the deeper into the topology the more and we may not have solid statistical support for diel-state for the many of the ancestral nodes. Second, by counting traits on ancestral nodes we run into the problem of over counting, the more the nodes, the more effective number of "samples" in that clade. The analyses we could do may not address the exact question but do point to the same trend. Specifically, we did examine something similar using extant taxa that had a single value for opsins at each tip. We checked if there were any differences in overall opsin counts with diel-niche and found no significant differences for all opsin counts between nocturnal and diurnal taxa. We suspect this is because of the poor reconstruction of UV and Blue because of missing values. However, among LW opsins where the reconstruction is more robust, we found LW opsins counts were higher in diurnal taxa than nocturnal taxa. (mean N = 1.139, mean D = 1.277, p-value= 0.033). To do our best to appease the reviewer, we have also moved this claim to the discussion because of ambiguity in verifying the claim quantitatively. (New text: Lines: 121-122, 217-218). We've done our best to address this criticism with the tools currently available, without overstating our results.

Comments: lines 148,150,154: *p-values are not a measure of effect size, thus phrases like "highly significant" and "moderately significant" are not appropriate. These should simply be phrased as significant or not, then effect sizes compared.*

Response: Changed text to reflect differences between significance and effect size.

Comments : lines 158-169: *This is the only section that I find troublesome. Upon reading, it sounds as if the authors attempted multiple evolutionary tests until they found one that supported their conclusions. For example the PAML models found "no significant signature of positive selection" (line 159) and the "LRT tests were not significant" (lines 161-2). Despite describing the LRT tests as "fairly*

conservative" (lines 162-3), subsequent, "more sensitive" (line 163,168) tests were applied in HYPHY. These new tests then report results consistent with the hypothesis. My point here is that if the HYPHY models are "tailored to detect specific kinds of evolutionary scenarios" (line 168-9), then please provide some kind of support/justification or description of why the reader should trust these results more than those from PAML, considering their conclusions are conflicting.

Response: Added justification for why these models will not identify positive selection in opsins and the need for using more sensitive models.

New text: "The site models test if any sites have different dN/dS rates across all branches. The analyses found no significant sites under positive selection across all branches showing that if diel-niche is not considered, there is no underlying signal of elevated dN/dS rates. The branch-site models, which allow partition by diel-niche, did identify amino acid sites that were under a higher dN/dS in diurnal species in UV, Blue and LW opsins, however it was not able to classify them as being under positive selection. This is expected in opsins because they have a low dN/dS rates overall⁴⁸, with $\omega < 0.5$, and the null models fixes dN/dS rates to $\omega = 1$, preventing it from classifying it as significant positive selection. Instead, we used more sensitive tests like the Contrast-FEL⁴⁹ and MEME models⁵⁰ from HyPhy to verify the branch site models (Table S4). These models look for differences in dN/dS rates in different sites among different branches, as opposed to looking for rates greater than one and thus are more appropriate for opsins. They are more recent implementations and are designed to be more sensitive for detecting selection since they are tailored to detect specific kinds of evolutionary scenarios⁴⁹⁻⁵¹."

Comments: lines 177-187: *this entire section describes methods and not results. I recommend placing in the 'Methods' section.*

Response: Moved the section to methods.

Comments: lines 178-179: *Should the reader be concerned that only 6 transmembrane domains were identified in most opsins other than the UV opsins? Having 7 transmembrane domains is a signature of these GPCRs.*

Response: This is only because the final alignment is shorter than the individual sequences to ensure more accurate mapping, the individual blue and LW opsin sequences had 7 transmembrane domains.

Comments: lines 197-204: *this section does not include any results but rather interpretation, I recommend moving to the "Discussion" section.*

Response: Agree and moved the lines to discussion

Comments: lines 240-241: *Again, see my comment above regarding the justification of using the Hyphy results rather than PAML results.*

Response: Agree and gave a more appropriate justification for why the HyPhy models are better for opsins.